# Current-driven writing process in anti-ferromagnetic Mn$_2$Au for memory applications

S. Reimers[1,6], Y. Lytvynenko [1,2,6], Y. R. Niu[3], E. Golias [3], B. Sarpi[4], L. S. I. Veiga[4], T. Denneulin [5], A. Kovács [5], R. E. Dunin-Borkowski[5], J. Bläßer[1], M. Kläui [1] & M. Jourdan [1] ✉

Current pulse driven Néel vector rotation in metallic antiferromagnets is one of the most promising concepts in antiferromagnetic spintronics. We show microscopically that the Néel vector of epitaxial thin films of the prototypical compound Mn$_2$Au can be reoriented reversibly in the complete area of cross shaped device structures using single current pulses. The resulting domain pattern with aligned staggered magnetization is long term stable enabling memory applications. We achieve this switching with low heating of ≈20 K, which is promising regarding fast and efficient devices without the need for thermal activation. Current polarity dependent reversible domain wall motion demonstrates a Néel spin-orbit torque acting on the domain walls.

Novel storage concepts in spintronics based on antiferromagnets (AFMs) propose to encode information by the direction of the alignment of the staggered magnetization or Néel vector[1–4]. This approach takes advantage of the intrinsically fast THz dynamics of AFMs[5] and their stability against external magnetic fields (see Ref. [6]). The interest in such concepts has grown strongly with the prediction of a so called Néel spin-orbit torque (NSOT) in metallic collinear AFMs with a specific symmetry, which is expected to rotate the spins of the AFM sublattices in a perpendicular orientation with respect to a driving bulk current[7].

Correspondingly, within the first generation of experiments on both known compounds with the required structure and AFM ordering, CuMnAs and Mn$_2$Au, current pulse induced reversible resistance modifications were reported as evidence for NSOT driven Néel vector rotation[8–13]. However, it later became clear that alternative mechanisms such as current pulse induced heat effects[14,15], electromigration[16], and rapid quenching induced structural and magnetic modifications[17] can result in resistance modifications similar to the ones reported as well. Additionally, external strain supported Néel vector manipulation in Mn$_2$Au was demonstrated[18,19]. Thus, microscopic investigations showing the intended current driven alignment of the staggered magnetization directly are essential. Up to now, for CuMnAs as well as

for Mn$_2$Au, only minor current induced modifications were observed, i.e., only a small fraction of the antiferromagnetic domains switched with limited reversibility and stability[9,20]. This is neither sufficient regarding the identification of the switching mechanism as local inhomogeneities are involved nor does it demonstrate the applicability of current induced Néel vector switching for memory devices.

Furthermore, microscopic imaging provides direct insights into the current driven mechanisms of Néel vector reorientation[21]. This recently attracted renewed interest, as in NiO/Pt bilayers thermo-magnetoelastic coupling effects were observed, which act in a very similar way as spin-orbit torques on the Néel vector[22]. Thus, the experimental demonstration of a current induced NSOT is important.

Here we show current pulse induced complete, remanent, and reversible Néel vector switching of Mn$_2$Au(001) thin films patterned in cross-structures, which is compatible with spintronics applications such as magnetic random access memory (MRAM). A current polarity dependence demonstrates an NSOT acting on AFM domain walls. Studying different pulse lengths, we show that in contrast to related experiments on Mn$_2$Au[12], for our epitaxial thin films thermal activation is not necessary for switching, which is important for fast and energy efficient memory applications regarding potential ultrafast applications.

[1]Institut für Physik, Johannes Gutenberg-Universität Mainz, Mainz, Germany. [2]Institute of Magnetism of the NAS of Ukraine and MES of Ukraine, Kyiv, Ukraine. [3]MAX IV Laboratory, Lund, Sweden. [4]Diamond Light Source, Chilton, Didcot, Oxfordshire, UK. [5]Ernst Ruska-Centre for Microscopy and Spectroscopy with Electrons, Forschungszentrum Jülich, Jülich, Germany. [6]These authors contributed equally: S. Reimers, Y. Lytvynenko. ✉e-mail: jourdan@uni-mainz.de

## Results

Mn$_2$Au is a metallic antiferromagnet with a high Néel temperature above 1000 K[23]. The compound has a tetragonal crystal structure with two equivalent ⟨110⟩ easy axes in the (001)-plane and a strong out-of-plane magnetic anisotropy.

We investigate epitaxial Mn$_2$Au(001) thin films with a thickness of 45 nm grown on an epitaxial double buffer layer of 13 nm of Ta(001) on 20 nm of Mo(001) on MgO(001) substrates (see Supplementary Information). The samples are capped with 2 nm of SiN$_x$ and patterned by optical lithography and Argon ion beam etching.

The largest NSOT is expected for the current directions parallel and antiparallel to the Néel vector[7,24]. Correspondingly, we patterned a Mn$_2$Au(001) thin film in a cross structure oriented parallel to the easy ⟨110⟩ directions, which allows to send current pulses both parallel and perpendicular to the axis along which the Néel vector is aligned. We obtain a repeatable complete switching of the Néel vector orientation in the central area (10 μm × 10 μm) of the cross by applying current pulses alternating between the two orthogonal directions. Figure 1 shows X-ray magnetic linear dichroism - photoelectron emission microscopy (XMLD-PEEM) images obtained after subsequent pulses, in which the 90° reorientation of the Néel vector shows up as alternating dark and white contrast of the central area.

For the investigation of the role of current heating induced thermal activation[12], we compare current pulse trains with different numbers and lengths, ranging from a train of 100 pulses with a lengths of 1 ms down to a single 10 μs pulse. The panels **a** and **b** of Fig. 1 show an example of complete switching after a train of 100 current pulses with a length of 1 ms each (off time after each pulse 10 ms). The panels **c** and **d** show very similar switching obtained after a single bipolar current pulse with a length of 10 μs only. Although only a single pulse instead of 100 pulses was used and the pulse length was 100 times shorter, the required current densities to obtain complete Néel vector rotation are very similar with $J_{1ms} ≈ 2.6 × 10^{11}$ A/m$^2$ and $J_{10μs} ≈ 3.0 × 10^{11}$ A/m$^2$ (and $J_{100μs} ≈ 2.7 × 10^{11}$ A/m$^2$, not shown). These current densities and pulse lengths result in significantly different temperature increases of the Mn$_2$Au thin film due to ohmic heating with $ΔT(1 ms) ≈ 70$ K, $ΔT(100 μs) ≈ 45$ K, and $ΔT(10 μs) ≈ 20$ K, as determined from ex-situ reference resistance measurements during application of current pulses (see Supplementary Information).

Next we investigate how the switched area of the sample depends on the current density. We patterned a single stripe along an easy [110] direction of a Mn$_2$Au thin film and applied 1 ms current pulses with increasing current density, as shown in Fig. 2. The single stripe geometry ensures a homogeneous distribution of the current density. The transition from the beginning of Néel vector reorientation to complete switching with increasing pulse current density occurs within a range of ≈20% of the maximum required current.

The left panel of Fig. 2 shows the initial domain configuration, in the as grown state after patterning. It is characterised by a preferential alignment of the Néel vector parallel to the stripe, which originates from patterning induced anisotropy[25]. With increasing amplitude of the current pulses (panels from left to right in Fig. 2), the first reorientation of the Néel vector appears in some central regions of the stripe indicated by yellow circles with a current density of $J_{1ms} = 2.46 × 10^{11}$ A/m$^2$. With further increasing current, the switched area increases homogeneously distributed over the patterned area, consistent with the absence of patterning induced current inhomogeneities. In the same experiment with an MBE grown Mn$_2$Au(001) thin film[26] and shorter current pulses (20 μs) very similar results were obtained, showing the robustness of the switching properties.

In order to identify if the switching is driven by NSOT, we investigated the current direction and polarity dependence of the Néel vector reorientation. This allows us to compare the experimental results of changes to the AFM domains and domain wall positions to theoretical predictions for NSOT[7,24]. Current pulses were applied along a hard [100] direction of a Mn$_2$Au(001) thin film using a single patterned stripe aligned along this direction. In this geometry, no net alignment of the Néel vector was obtained in the central region of the patterned stripe. However, we observed a partial current polarity dependent reversible reorientation. Some of these regions which switch for positive current polarity from vertical to horizontal alignment are indicated by the red circles in Fig. 3. Examples of other regions which switch in the opposite direction are indicated by yellow circles. The green circles in panel **f** show regions, in which the current was flowing preferentially along an easy ⟨110⟩-direction resulting in a local alignment of the Néel vector corresponding to the behavior discussed above (Figs. 1 and 2).

Finally, we determine the change of the sample resistance associated with the Néel vector reorientation discussed above. For this, an 8-terminal shaped device as shown in Fig. 4b was patterned. We applied bipolar current pulses alternating between the two orthogonal easy ⟨110⟩ directions, as done previously for the 4-terminal cross structures shown in Fig. 1. With the 8-terminal geometry, both the longitudinal $R_{long}$ as well as the transversal $R_{trans}$ sample resistance can be probed after each current pulse as shown in Fig. 4. Pulsing with the same current densities as required for the observation of the Néel vector reorientation by X-PEEM, we observed alternating longitudinal (a) as well as transversal (see Supplementary Material) resistance

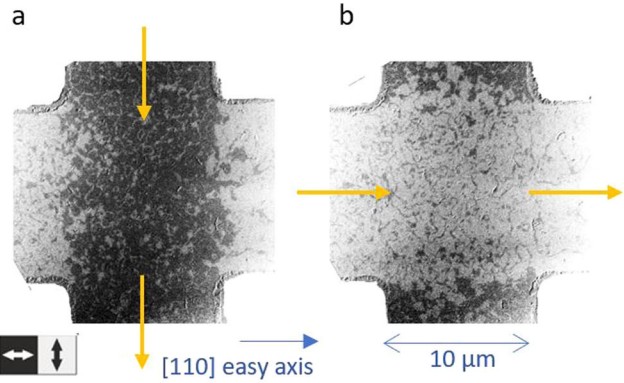
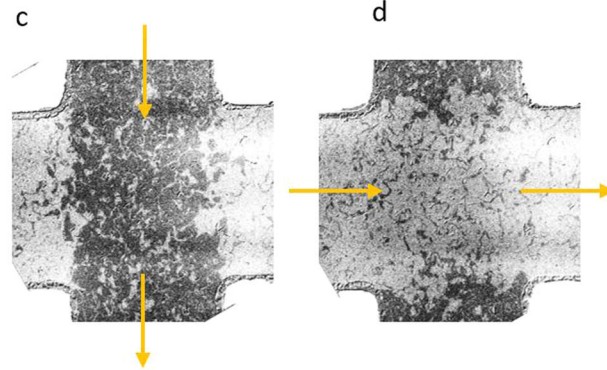

a  b  [110] easy axis  10 μm

c  d

**Fig. 1 | XMLD-PEEM images of reversible Néel vector reorientation, with current parallel easy axes.** The images show the orientation of the Néel vector of Mn$_2$Au(001) thin films after sending current pulses with different length and direction (yellow arrows) through a patterned cross structure oriented parallel to the easy ⟨110⟩ directions. The dark regions correspond to a horizontal, the bright regions to a vertical alignment of the Néel vector as indicated below panel **a** by the double arrows. **a, b** After 100 pulses of 1 ms length each with a current density of $J = 2.6 × 10^{11}$ A/m$^2$. **c, d** After 1 bipolar pulse of 10 μs length with a current density of $J = 3.0 × 10^{11}$ A/m$^2$.

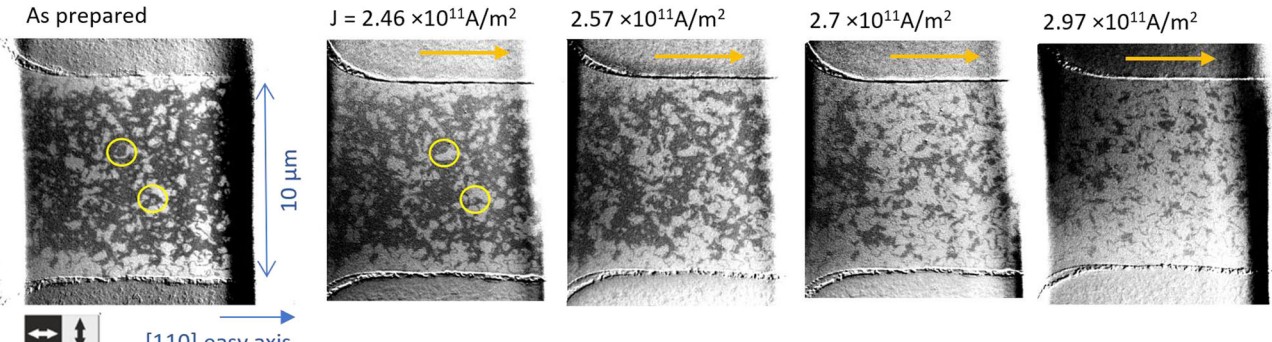

**Fig. 2 | XMLD-PEEM images of Néel vector reorientation, with current along easy axis.** The images show the orientation of the Néel vector of Mn$_2$Au(001) thin films as grown and after sending current pulses (length 1 ms) with increasing amplitude along an easy ⟨110⟩ direction through a patterned stripe structure. The yellow arrows indicate the current directions.

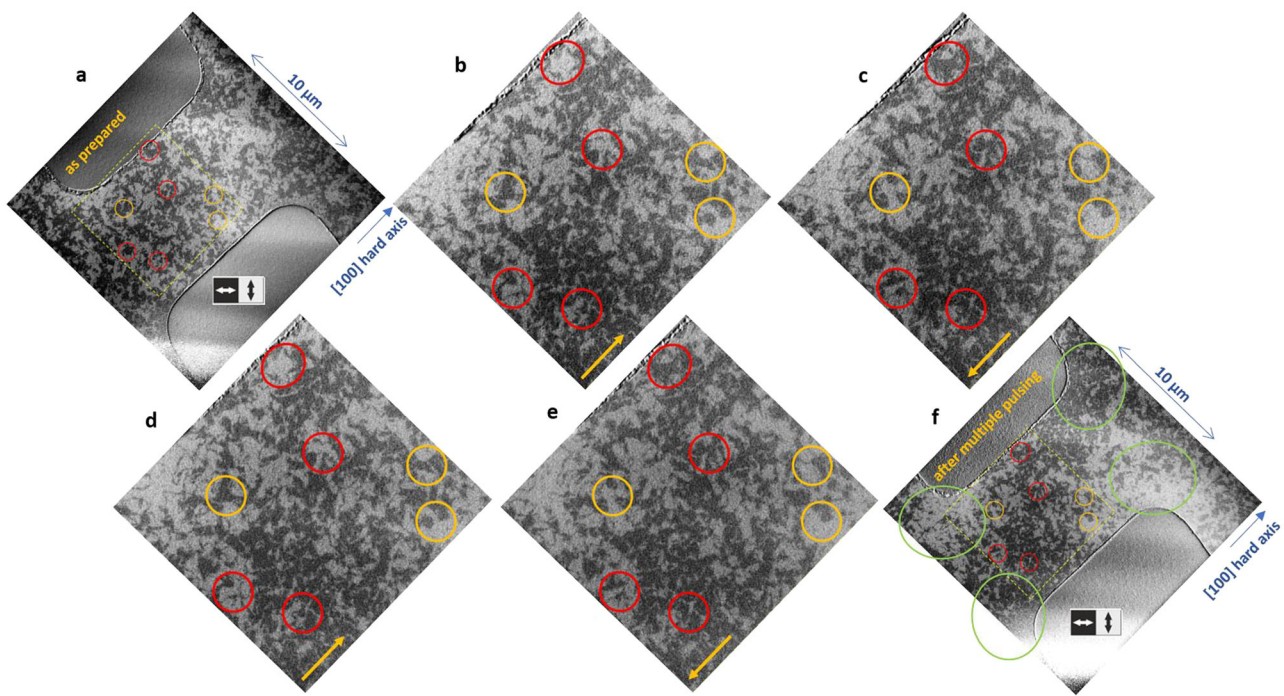

**Fig. 3 | XMLD-PEEM images of Néel vector reorientation, with current parallel hard axis.** The images show the orientation of the Néel vector of Mn$_2$Au(001) thin films as grown **a** and after sending current pulses (length 1 ms, amplitude $J = 3.6 \times 10^{11}$ A/m$^2$) with alternating polarity indicated by yellow arrows along the hard [100] direction through a patterned stripe structure with panels **b** to **e** showing the region indicated by the yellow dashed square in panel **a** with increased magnification. **f** Shows the full field of view after the last current pulse.

values. Both magnetoresistance signals are as expected of similar magnitude, slightly inhomogeneous Néel vector alignment in the central area of the 8-terminal cross can explain the remaining differences.

We obtained a maximum $\Delta R_{\text{long}}/R_{\text{long}} \simeq 1 \times 10^{-3}$, which is consistent with the low temperature anisotropic magnetoresistance value of AMR$_{\text{Mn}_2\text{Au}} \simeq -1.5 \times 10^{-3}$, which we obtained previously by aligning the Néel vector with a 50 T magnetic field pulse[27]. Furthermore, the negative AMR of Mn$_2$Au (i.e., $\rho_\perp > \rho_\parallel$) is consistent with the larger longitudinal resistance in Fig. 4a associated with pulse current direction $I_{\text{pulse1}}$ (blue data points), if the Néel vector is aligned perpendicular to the current direction as expected for an NSOT acting.

Finally, consistent with the X-PEEM measurements, the electrical signal shows no sign of decay as shown in the Supplementary Information.

## Discussion

For potential spintronics applications such as antiferromagnetic random access memory (MRAM), long term stability of the Néel vector aligned states is required. In this framework our Mn$_2$Au(001) thin films are very suitable as the current pulse induced AFM domain configurations perfectly fulfill this requirement. We were able to demonstrate by XMLD-PEEM that even four months after the current pulse Néel vector alignment shown in Fig. 2, the aligned AFM domain configuration was preserved (see Supplementary Information). Additionally, already single current pulses are able to reorient the Néel within the central area of device cross shaped structures completely and reversibly, i.e., again fulfilling the requirements for applications.

For spintronics, the size of the electrical read-out signal associated with Néel vector reorientation is of major importance. Our work confirms that the maximum obtainable signal from AMR is relatively small so that in antiferromagnetic spintronics other read-out

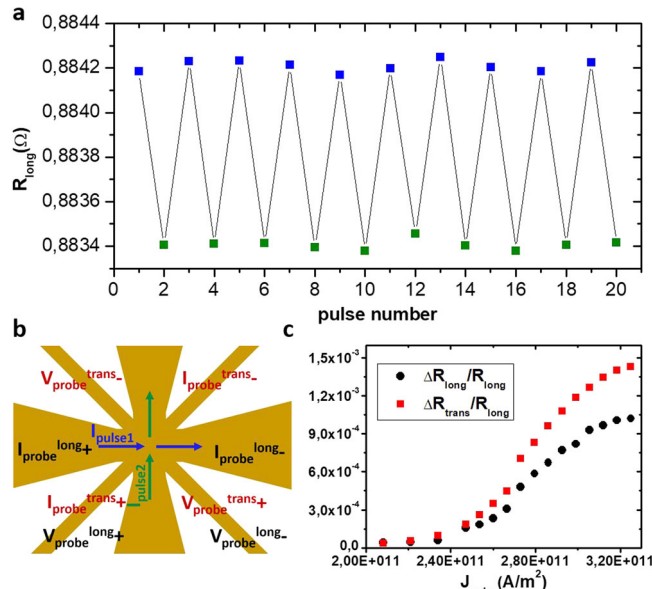

**Fig. 4 | Switching induced anisotropic magnetoresistance. a** Shows the alternating longitudinal resistance of a $Mn_2Au$ thin film patterned in the geometry shown in **b**. The longitudinal resistance was measured after each 1 ms current pulse with $J_{pulse} = 3 \times 10^{11}$ A/m$^2$ applied alternately in perpendicular directions as indicated by the blue and green arrows in **b**. The color of the data points corresponds to the arrows indicating the current direction. **c** Shows the dependence of the longitudinal $\Delta R_{long}$ as well as of the transverse $\Delta R_{trans}$ resistance changes on the pulse current density $J_{pulse}$.

mechanisms, for example via an adjacent strongly exchange-coupled ferromagnetic layer as discussed in reference[28], need to be considered. Furthermore, our small electrical read-out signals are clearly distinguishable from the current pulse induced large resistance modifications previously obtained investigating various non-NSOT related metals as mentioned in the introduction.

Regarding the physical mechanism of the current pulse induced Néel vector reorientation, we compare our experiments with theoretical predictions assuming an NSOT, which is expected to be largest for current direction parallel to the sublattice magnetization and zero for current perpendicular to it[7,24]. Fully consistent with this prediction, we observe a current induced Néel vector orientation perpendicular to the pulse direction, if the current flows parallel to an easy $\langle 110 \rangle$-axis of $Mn_2Au$ (Figs. 1 and 2). However, also thermomagnetoelastic coupling effects driven by anisotropic strain due to current heating can generate this type of Néel vector reorientation[22], as the Néel vector prefers alignment along an elongated $\langle 110 \rangle$ direction[19]. In principle, thermomagnetoelastic coupling and NSOT can cooperate for stripe and cross geometries aligned along $\langle 110 \rangle$ directions.

To study the action of the NSOT independently from this potentially additional contribution, we discuss the reversible current polarity dependence of the Néel vector reorientation, which cannot appear due to any thermally driven effect which acts identically for both current directions. For current pulses parallel to a hard $\langle 100 \rangle$-direction, the strongest NSOT is expected to act on the AFM domain walls, as only there the Néel vector is aligned parallel to the current. In this case the NSOT is expected to shift the domain wall reversibly with inverted current direction as observed previously in CuMnAs(001) thin films[21]. As discussed above and shown in Fig. 3, we demonstrated such a current polarity dependent reorientation of the Néel vector as well. In $Mn_2Au$, a relatively large number of domains shows reversible modifications, but also in our compound the domain walls are shifted by distances of only $\approx 1\,\mu m$. This indicates that mechanisms such as domain wall pinning are competing with the NSOT acting on the

domain walls. Nevertheless, the reversible current polarity dependent Néel vector reorientation is direct evidence for the action of a current induced NSOT.

In contrast to domain wall pinning, the potential barrier for Néel vector reorientation which is typically considered in theoretical work[7,29] is the in-plane magnetocrystalline anisotropy ($\approx 1.8\,\mu eV$ per formula unit for $Mn_2Au(001)$[28]). In this framework a key role of thermal activation was previously reported based on investigations of granular $Mn_2Au$ thin films[12]. However, as shown e. g. by the transmission electron microscopy image in the Supplementary Information, our $Mn_2Au(001)$ thin films are highly epitaxial without morphological features on the typical length scale of the AFM domains, so that independent switching of morphological grains can be excluded. This strongly suggests that the reorientation occurs via domain wall motion.

Independent of the physical origin of the potential barriers to be overcome, it is important for spintronics applications that the current induced Néel vector reorientation does not depend on slow thermal activation processes. Here, the required current density for full Néel vector reorientation using a single 10 µs current pulse was only $\approx 10\%$ larger than required for a pulse train of 100 current pulses with 1 ms pulse length each. This indicates that the current pulse induced force is at least of the same order of magnitude as opposing forces e.g., due to domain wall pinning.

We have shown that metallic antiferromagnets, specifically epitaxial $Mn_2Au(001)$ thin films, are finally able to fulfill one of the major early promises of antiferromagnetic spintronics: it is possible to write long term stable information by single current pulses into macroscopic areas of device structures. Regarding the switching mechanism, while thermomagnetoelastic contributions might be present, we identify a Néel spin-orbit torque acting on the domain walls leading to reversible motion. Furthermore, we have shown that thermal activation processes are not essential for current induced Néel vector reorientation, thereby enabling fast and energy efficient switching. Thus, epitaxial $Mn_2Au(001)$ thin films fulfill all requirements for memory applications regarding the writing and storage of data using a single antiferromagnetic layer.

## Methods

All experimental data shown in this manuscript were obtained investigating $Mn_2Au(001)(45\,nm)$ thin films grown epitaxially on Ta(001) (13 nm)/Mo(001)(20 nm) double buffer layers on MgO(100) substrates. All layers were deposited by magnetron sputtering by the process described in detail in ref. 28. The samples were capped with 2 nm of polycrystalline $SiN_x$ to protect them from oxidation. Optical lithography and ion beam etching were used to pattern the films into the cross and stripe structures shown in this manuscript.

Antiferromagnetic domain imaging was performed by combining photoemission electron microscopy with x-ray magnetic linear dichroism (XMLD-PEEM) at the Mn $L_{2,3}$ absorption edge at the PEEM endstations at beamline MAXPEEM at MAX IV, and beamline I06 at Diamond Light Source. The XMLD effect at the Mn $L_{2,3}$ absorption edge in $Mn_2Au$ was established in previous work. For x-ray polarisation along a $Mn_2Au$ $\langle 110 \rangle$ direction, the Mn $L_{2,3}$ XMLD spectrum shows a minimum and a maximum located at the absorption edge $E_{max}$ and at 0.8 eV below the edge. At MAXPEEM, the x-ray beam has normal incidence at the sample surface. The XMLD-PEEM images obtained at this beamline are the asymmetry images of images taken with two orthogonal x-ray polarisations along the $\langle 110 \rangle$ directions and energy of $h\nu = E_{max}$-0.8 eV, as this provides maximum contrast and minimum sensitivity to morphological features. At I06, the x-ray beam is incident at a grazing angle of 16°. XMLD-PEEM images measured there are the asymmetry of images with photon energies of $h\nu = E_{max}$ and $h\nu = E_{max} - 0.8$ eV for fixed in-plane polarisation along a $\langle 110 \rangle$ direction.

In-situ electrical manipulation was performed using the pulse functions of Keithley2601B-PULSE (at MAX IV) and Keithley 2461 (at Diamond) sourcemeters, integrated into the X-PEEM setup.

The ex-situ resistance measurements (Fig. 4) were performed using a Keithley 6220 precision current source with a probe current of 50 µA and a Keithley 2182A Nanovoltmeter in Delta mode averaging over 200 measurements to obtain one data point. For automatising the pulse (Keithley 2430 Pulse Source Meter) - probe sequence an Agilent 34970A Switch Unit was used.

For the quantification of the current pulse induced sample heating, we used a lithographic pattern, which additional to the cross shown in Fig. 1 contains four thin leads enabling longitudinal resistance measurements of the central cross area with a 4-probe technique (see Fig. 1 of ref. 11). The maximum resistance obtained at the end of the current pulses with different lengths and amplitudes is shown in the Supplementary Information. By comparing this high pulse current sample resistance with the above room temperature linear extrapolated temperature dependent sample resistance $R(T)$ obtained from a low current measurement in a cryostat, we can deduce the current pulse induced temperature raise. These measurements were performed outside the PEEM at ambient conditions using the same Keithley 2601B-PULSE Source Meter as for pulsing in the PEEM. The time dependent sample resistance during the current pulses was measured with an oscilloscope and using the SENSE inputs of the Source Meter obtaining consistent results.

## Data availability

The raw data of the transport measurements generated in this study have been deposited in the Zenodo database under accession code https://doi.org/10.5281/zenodo.7467238 [https://zenodo.org/record/7467238#.Y_h6mHbMKKs].

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

## Acknowledgements

We acknowledge funding by the Deutsche Forschungsgemeinschaft (DFG, German Research Foundation)—TRR 173—268565370 (project A05 (M.J.), with contribution from A01 (S.R., M.K.)), by the Horizon 2020 Framework Program of the European Commission under FET-Open Grant No. 863155 (s-Nebula) (M.K.), by EU HORIZON-CL4-2021-DIGITAL-EMERGING-01-14 programme under grant agreement No. 101070287 (S.R., M.K.), and by the TopDyn Center (M.K.). B.S. is supported by the UK Engineering and Physical Science Research Council (Grants No. EP/V029258/1). We acknowledge MAX IV Laboratory for time on beamline MAXPEEM under Proposal 20210863 (M.J.), and Diamond Light Source for time on beamline I06 under proposal MM30141-1 (M.J.). Research conducted at MAX IV, a Swedish national user facility, is supported by the Swedish Research council under contract 2018-07152, the Swedish Governmental Agency for Innovation Systems under contract 2018-04969, and Formas under contract 2019-02496. The STEM investigations were funded by the European Union's Horizon 2020 Research and Innovation Programme under grant agreement 856538 (project "3D MAGIC") (M.K. and R.E.D.-B.).

## Author contributions

S.R., Y.L., and M.J. wrote the paper, prepared the samples and performed the XMLD-PEEM investigations. Y.R.N and E.G. supported the XMLD-PEEM investigations at MAX IV, B.S. and L.S.I.V. at Diamond, and A.K. at SLS. S.R. and J.B. measured the Néel vector induced resistance changes, Y.R.N. measured the pulse heating induced resistance changes. T.D., A.K., and R.E.D.-B. contributed the STEM investigations. M.K. contributed to the discussion of the results and provided input, M.J. coordinated the project.

## Funding

## Competing interests

The authors declare no competing interests.
