## [Peer Review File · Nature Communications]

Reviewers' Comments:

Reviewer #1:

None

Reviewer #2:

Remarks to the Author:

The manuscript deals with the experimental confirmation of the mechanism of Néel spin orbit torque in Mn₂Au. Some of the authors are pioneers in the field of current induced switching of the Néel vector in metallic antiferromagnets with oppositely broken inversion symmetry, notably Mn₂Au. This included the experimental confirmation via XMLD-PEEM of current induced Neel vector witching in Mn₂Au in their 2019 PRB 99, 140409(R).

The work presented here is a continuation of the work published in 2019 with respectable but incremental progress. Granted the effects a more pronounced now and hence more convincing, but they have no significant qualitative new component compared to the bulk of work (Refs.[6,11,18,26]) already published by some of the authors of the manuscript under consideration. Although the manuscript is solid experimental work, I don't see how it meets the originality standards of Nature Communications.

Minor criticism

Yellow circles should also be explained in captions of Fig. 2?

Yellow and red circles should also be explained in captions of Fig. 3?

Reviewer #3:

Remarks to the Author:

In this work, Lytvynenko et al. reported current-driven switching of antiferromagnetic moments in Mn₂Au. Current pulses along the easy axes $\langle 110 \rangle$ can reorientate the Néel vector of Mn₂Au to the perpendicular direction of electric current, the reorientation process can be clearly observed by XMLD-PEEM images. By investigating the dependence of critical switching current density on the numbers and widths of current pulses and the current polarity-dependent domain switching, the important role of NSOT is confirmed. This work is quite helpful to understand the mechanism of NSOT and can be considered for Nature Communications if the following questions are well addressed.

1) This work discusses the contribution of NSOT and thermomagnetoelastic strain to the switching of Mn₂Au. Whether the magnetoelastic constant of Mn₂Au is positive or negative? Does the thermomagnetoelastic strain cooperate to or cancel NSOT?

2) The main conclusion is supported by XMLD-PEEM images. Some independent electrical measurements will make this work more complementary. For example, in Ref. 19 the reorientation of Mn₂Au induced by positive and negative can be electrically detected by Hall resistance. Similar measurements can be performed for Mn₂Au. The simultaneous electrical signals and domain images are quite helpful for the quality of the data.

3) The retention measurement of Mn₂Au is important for memory application. And the cycling of current-induced reversible switching is also important. Authors performed long term stability of only one switched state in Fig. S2. I suggest authors to perform reversible switching with long delay time (for example 1h).

4) Some other works relevant to NSOT and spin current generation of Mn₂Au are suggested to be referred, such as Nat. Mater. 18, 931 (2019) and Nat. Mater. 20, 800 (2021).

Mainz, 21.12.2022

Dear Reviewers,

We thank the reviewers for their careful reading of our manuscript and their valuable comments. Please find below citations of the reviewer reports (printed in blue), together with our corresponding reply. Additionally, all changes to the manuscript are listed at the end of this reply letter.

As we added XMLD-PEEM data and resistance measurements in reply to the reviewer reports, a new author was added to the manuscript (J. Bläßer). Furthermore, we changed the order of the two equally contributing authors Y. Lytvynenko and S. Reimers, so that S. Reimers, who devised and led the experiments for all the added results in the revised version, is now first author.

Comment by Reviewer #2:

The manuscript deals with the experimental confirmation of the mechanism of Néel spin orbit torque in Mn₂Au. Some of the authors are pioneers in the field of current induced switching of the Néel vector in metallic antiferromagnets with oppositely broken inversion symmetry, notably Mn₂Au. This included the experimental confirmation via XMLD-PEEM of current induced Neel vector witching in Mn₂Au in their 2019 PRB 99, 140409(R).

The work presented here is a continuation of the work published in 2019 with respectable but incremental progress. Granted the effects a more pronounced now and hence more convincing, but they have no significant qualitative new component compared to the bulk of work (Refs.[6,11,18,26]) already published by some of the authors of the manuscript under consideration. Although the manuscript is solid experimental work, I don't see how it meets the originality standards of Nature Communications.

Reply:

Regarding the relation to our and to other previous work, we would like to emphasize the statement from our manuscript, "Up to now, for CuMnAs as well as for Mn₂Au, only minor current induced modifications were observed, *i. e.* only a small fraction of the antiferromagnetic domains switched with limited reversibility and stability [9, 18].", by showing below a direct comparison of the results from our previous work (Fig. 3 of Ref. 18) and the reversible switching of our current work.

Our previous work:

FIG. 3. Difference images calculated from XMLD-PEEM asymmetry images obtained before and after the subsequent application of current pulse trains with $I = 1.25 \times 10^7$ A/cm² along the easy-axis directions [110] and [1 $\bar{1}$ 0]. The current direction for each panel (a)–(d) is indicated by the blue and red arrows. The yellow circles indicate obvious examples of reversibly switching AFM domains, whereas the red circles indicate examples of domains switching irreversibly.

Current manuscript:

FIG. 1. XMLD-PEEM images of reversible Néel vector reorientation, with current parallel easy axes. The images show the orientation of the Néel vector of Mn₂Au(001) thin films after sending current pulses with different length and direction (yellow arrows) through a patterned cross structure oriented parallel to the easy (110) directions. The dark regions correspond to a horizontal, the bright regions to a vertical alignment of the Néel vector as indicated below panel **a** by the double arrows. **a, b**: After 100 pulses of 1 ms length each with a current density of $J = 2.6 \times 10^{11}$ A/m². **c,d**: After 1 bipolar pulse of 10 μ s length with a current density of $J = 3.0 \times 10^{11}$ A/m².

Ref. 18 represents the first direct imaging of any current induced Néel vector manipulation in Mn₂Au, but only a few small grains were switched reversibly. The demonstration of technologically relevant reliable large area switching was missing. Additionally, the previous work did not allow to draw conclusions regarding the switching mechanism. In this manuscript, we provide evidence for a current pulse induced NSOT by (i) showing a current polarity dependent Néel vector reorientation; (ii) studying the influence of heating; and (iii) showing that the Néel vector is rotated perpendicular to the applied current. Furthermore, we added electrical measurements in the revised version of the manuscript, which show the anisotropic magnetoresistance (AMR) associated with the Néel vector reorientation. By this, we are now able to clearly distinguish our electrical read-out signals from the current pulse induced large resistance modifications previously obtained investigating various non-NSOT related metals as mentioned in the introduction of our manuscript.

We are convinced that our manuscript presents novel and unique results on current induced Néel vector reorientation in antiferromagnets, which are of high technological relevance and which clarify that NSOT is a driving mechanism for current-induced switching in Mn₂Au.

Referee #3:

1) This work discusses the contribution of NSOT and thermomagnetoelastic strain to the switching of Mn₂Au. Whether the magnetoelastic constant of Mn₂Au is positive or negative? Does the thermomagnetoelastic strain cooperate to or cancel NSOT?

Our just published Ref. [19] shows that elongating one of the easy <110> directions by external strain results in a preferred alignment of the Néel vector along the direction of elongation (without any current induced manipulation).

We now extend the discussion of this issue from the previous version of our manuscript where we wrote “However, also thermomagnetoelastic coupling effects driven by anisotropic strain due to current heating can generate this type of Néel vector reorientation [22].”, by adding:

“..., as the Néel vector prefers alignment along an elongated <110> direction [19]. In principle, thermomagnetoelastic coupling and NSOT can cooperate for stripe and cross geometries aligned along <110> directions.

2) The main conclusion is supported by XMLD-PEEM images. Some independent electrical measurements will make this work more complementary. For example, in Ref. 19 the reorientation of Mn₂Au induced by positive and negative can be electrically detected by Hall resistance. Similar measurements can be performed for Mn₂Au. The simultaneous electrical signals and domain images are quite helpful for the quality of the data.

Thank you very much for this remark, which resulted in very valuable new insights. We performed the requested additional resistance measurement (ex-situ transverse and longitudinal resistance measurements using the pulse same current densities for switching) and added Fig. 4 and corresponding explanations to the manuscript. Please have a look at the detailed list of changes of the manuscript given below.

3) The retention measurement of Mn₂Au is important for memory application. And the cycling of current-induced reversible switching is also important. Authors performed long term stability of only one switched state in Fig. S2. I suggest authors to perform reversible switching with long delay time (for example 1h).

Thank you very much for this remark, which relates our results to the decay of the resistance change associated with switching in other compounds. However, we did not observe any decay at all. During our XMLD-PEEM beam time, we imaged switched states several times with delays of one to several hours (food and night breaks) and never saw any relaxation of the AFM domain pattern. Regarding Fig. S2, we explicitly used it to demonstrated stability over several months.

Additionally, our meanwhile established pulse induced AMR measurements (please see remark (2) above) now allow us to investigate potential decay behaviour with a probing frequency of *seconds* (in contrast to PEEM, where obtaining an image typically takes 1 min). Again, we observed no relaxation behaviour, as shown now in Fig. S4 of the Supplementary Material.

4) Some other works relevant to NSOT and spin current generation of Mn₂Au are suggested to be referenced, such as *Nat. Mater.* 18, 931 (2019) and *Nat. Mater.* 20, 800 (2021).

Regarding strain supported Néel vector manipulation, we added in the introduction as proposed the above mentioned first reference (now [18] and an additional reference (now [19])). Regarding the second proposed reference about the antiferromagnetic spin Hall effect, which is a SOT effect observed at the *interface* between Mn₂Au and a ferromagnet, we feel that adding it would break the flow of our introduction about the *bulk* NSOT.

List of all changes in the revised version of our manuscript:

We added to the Introduction:

Additionally, external strain supported Néel vector manipulation in Mn_2Au was demonstrated [18,19]. (Both referenced added).

We added at the end of section “Results”:

Finally, we determine the change of the sample resistance associated with the Néel vector reorientation discussed above. For this, an 8-terminal shaped device as shown in panel **b** of Fig. 4 was patterned. We applied bipolar current pulses alternating between the two orthogonal easy $\langle 110 \rangle$ directions, as done previously for the 4-terminal cross structures shown in Fig. 1. With the 8-terminal geometry, both the longitudinal R_{long} as well as the transversal R_{trans} sample resistance can be probed after each current pulse as shown in Fig. 4. Pulsing with the same current densities as required for the observation of the Néel vector reorientation by X-PEEM, we observed alternating longitudinal (panel **a**) as well as transversal (see Supplementary Material) resistance values. Both magnetoresistance signals are as expected of similar magnitude, slightly inhomogeneous Néel vector alignment in the central area of the 8-terminal cross might explain the remaining differences.

We obtained a maximum $\Delta R_{\text{long}}/R_{\text{long}} \approx 1 \times 10^{-3}$, which is consistent with the low temperature anisotropic magnetoresistance value of $\text{AMR}_{\text{Mn}_2\text{Au}} \approx -1.5 \times 10^{-3}$, which we obtained previously by aligning the Néel vector with a 50 T magnetic field pulse [27]. Furthermore, the negative AMR of

Mn_2Au (i. e. $\rho_{\perp} > \rho_{\parallel}$) is consistent with the larger longitudinal resistance in Fig. 4 (panel **a**) associated with pulse current direction $I_{\text{pulse}1}$ (blue data points), if the Néel vector is aligned perpendicular to the current direction as expected for an NSOT acting. Furthermore, consistent with the X-PEEM measurements, the electrical signal shows no sign of decay as shown in the Supplementary Information.

We added in section “Discussion”:

For spintronics, the size of the electrical read-out signal associated with Néel vector reorientation is of major importance. Our work confirms that the maximum obtainable signal from AMR is relatively small so that other read-out mechanisms in antiferromagnetic spintronics, for example via an adjacent strongly exchange-coupled ferromagnetic layer as discussed in reference [29], need to be considered. Furthermore, our small electrical read-out signals are clearly distinguishable from the

FIG. 4. **Switching induced anisotropic magnetoresistance.** Panel **a** shows the alternating longitudinal resistance of a Mn_2Au thin film patterned in the geometry shown in panel **b**. The longitudinal resistance was measured after each 1 ms current pulse with $J_{\text{pulse}} = 3 \times 10^{11} \text{ A/m}^2$ applied alternately in perpendicular directions as indicated by the blue and green arrows in panel **b**. The color of the data points corresponds to the arrow indicating the current direction. Panel **c** shows the dependence of the longitudinal ΔR_{long} as well as of the transverse ΔR_{trans} resistance changes on the pulse current density J_{pulse} .

current pulse induced large resistance modifications previously obtained investigating various non-NSOT related metals as mentioned in the introduction.

However, also thermomagnetoelastic coupling effects driven by anisotropic strain due to current heating can generate this type of Néel vector reorientation [22], as the Néel vector prefers alignment along an elongated $\langle 110 \rangle$ direction [19]. In principle, thermomagnetoelastic coupling and NSOT can cooperate for stripe and cross geometries aligned along $\langle 110 \rangle$ directions.

We added in section “Methods”:

The ex-situ resistance measurements (Fig. 4) were performed using a Keithley 6220 precision current source with a probe current of $50\mu\text{A}$ and a Keithley 2182A Nanovoltmeter in Delta mode averaging over 200 measurements to obtain one data point. For automatising the pulse (Keithley 2430 Pulse Source Meter) - probe sequence an Agilent 34970A Switch Unit was used.

We added to the Supplementary Material:

Probing Néel vector reorientation by transverse resistance measurements

Electrical detection of current pulse induced Néel vector alignment is also possible via measurements of the transverse resistance or planar Hall-Effect. Considering the rather small electrical signal, this is more easy to measure than the longitudinal AMR, as there is no offset voltage. In Fig. S2, the measurement of the transverse resistance changes is shown in analogy to Fig. 4 of the main text.

Long term stable Néel vector alignment

...

Additionally, we demonstrate the absence of relaxation on short time scales, i. e. seconds, by measurements of the transverse resistance after a sequence of current pulse induced reversible switching. The data shown in Fig. S4 was obtained in the same way as for Fig. S2 (here with $J_{\text{pulse}} = 2.9 \times 10^{11} \text{ A/m}^2$), but we kept measuring the resistance after the last switching current pulse for 1.5 h. The raw data shown was obtained at a sampling rate of ≈ 2 data points per second. Also on this time scale, no relaxation of the resistance after the last switching pulse was observed.

Yours sincerely,

Martin Jourdan

FIG. 2. **S2: Switching induced planar Hall effect.** The measurements shows the alternating transverse resistance of a Mn_2Au thin film patterned in the geometry shown in Fig. 4, panel **b**. The transverse resistance was measured after each 1 ms current pulse with $J_{\text{pulse}} = 3 \times 10^{11} \text{ A/m}^2$ applied alternately in perpendicular directions as indicated by the blue and green arrows in Fig. 4, panel **b**. The color of the data points corresponds to the arrows indicating the current direction.

FIG. 4. **S4: Second time-scale stability of switched state.** Raw data (black data points) and averaged data points (over 200 raw data points) of the transverse resistance of a Mn_2Au thin film. Here, the transverse resistance was measured after each 1 ms current pulse with $J_{\text{pulse}} = 2.9 \times 10^{11} \text{ A/m}^2$ applied alternately in perpendicular directions followed by pure resistance measurements without pulsing. No relaxation was observed.

Reviewers' Comments:

Reviewer #1:

None

Reviewer #2:

Remarks to the Author:

Indeed, my debate between quantity and quality might be a bit too philosophical. I can agree that the transition from "minor current induced modifications" to "technologically relevant reliable large area switching" is a big enough quantitative change such that a new level of quality and thus novelty emerges from it. In addition, with new insights into the Neel spin-orbit torque switching mechanism, one can make the case that sufficient novel and unique results are presented.

Reviewer #3:

Remarks to the Author:

The authors performed new experiments and addressed my previous comments and questions well. I also found the other referee has some concerns concerning the novelty compared to the work published by the same group. I think the authors have answered this suitably, from the previous "hint" to the present "device", this is an obvious progress and provides clear evidence for the switching mechanism. Thus, I recommend the publication of this work as it is.